# Incidence of Common Herpesviruses in Colonic Mucosal Biopsies Following Hematopoietic Stem Cell Transplantation

**DOI:** 10.3390/microorganisms10112128

**Published:** 2022-10-27

**Authors:** Oleg V. Goloshchapov, Alexander N. Shvetsov, Alexey B. Chukhlovin, Anna A. Spiridonova, Maria D. Vladovskaya, Ludmila S. Zubarovskaya, Alexander D. Kulagin

**Affiliations:** 1R. Gorbacheva Memorial Research Institute of Pediatric Oncology, Hematology and Transplantation, Pavlov University, St. Petersburg 197022, Russia; 2Pediatric Research Clinical Center of Infectious Diseases, St. Petersburg 197022, Russia; 3St. Petersburg Pasteur Institute, St. Petersburg 197101, Russia

**Keywords:** hematopoietic stem cell transplant, Epstein–Barr virus, colon, mucosa, complications

## Abstract

Intestinal complications are common after allogeneic hematopoietic stem cell transplantation (allo-HSCT). However, only scarce data concern herpesvirus incidence in the colonic mucosa post-HSCT. Our purpose was to assess the frequency and clinical significance of cytomegalovirus (CMV), Epstein–Barr virus (EBV), human herpesvirus type 6 (HHV6), and herpes simplex virus (HSV) in the colonic mucosa post-HSCT. The study group included 119 patients of different ages, mostly with leukemias and lymphomas, subjected to allo-HSCT from haploidentical related (48%) or HLA-compatible donors (52%). In total, 155 forceps biopsies of the colonic mucosa were taken in cases of severe therapy-resistant intestinal syndrome post-HSCT. Most samples were taken from the descending, sigmoid, and transverse colon. Intestinal GVHD or local infections were assessed clinically and by histology. EBV, CMV, HSV, and HHV6 were tested in colonic mucosal lysates with commercial PCR assays. HSV was found in <8% of colonic samples, along with high HHV6 and CMV positivity (up to 62% and 35%, respectively) and a higher EBV incidence at 5–6 months post-HSCT (35%). For CMV and EBV, significant correlations were revealed between their rates of detection in blood and colonic mucosa (r = 0.489 and r = 0.583; *p* < 0.05). No significant relationships were found between the presence of herpesviruses and most patients’ characteristics. EBV positivity in colonic samples was correlated with delayed leukocyte and platelet recovery post-HSCT. Higher EBV frequency in the colonic mucosa was found in deceased patients (56% versus 21%, *p* = 0.02). The correlations among EBV positivity in the colon, lethality rates and delayed hematopoietic reconstitution suggest some relationship with systemic and local EBV reactivation post-transplant.

## 1. Introduction

Hematopoietic stem cell transplantation from allogeneic donor (allo-HSCT) is a common effective treatment option for different blood malignancies. However, intensive conditioning therapy prior to HSCT causes severe immune deficiency due to prolonged neutro- and lymphopenia, causing infectious complications. Moreover, post-HSCT expansion of the allogeneic donor cells may induce acute graft-versus-host disease (aGVHD). AGVHD of the gastrointestinal tract develops within 100 days after HSCT due to donor T cell-mediated immune lesions causing a severe intestinal syndrome with diarrhea, vomiting, and dyspeptic symptoms [1]. Regular immunosuppressive treatment post-HSCT also contributes to high rates of infectious complications. Intestinal bacterial microbiota in the patients is severely impaired after HSCT because of the massive antibiotic treatment over the post-transplant period [2]. Life-threatening intestinal disorders may require intensive care. Intestinal GVHD may also develop in later periods, manifesting as the so-called overlap syndrome, combining features of acute and chronic GVHD.

The impaired immune response post-transplant is caused by delayed recovery of T- and B cells providing an antiviral immune response [3]. For example, various herpesviruses are acquired by humans from their birth. Early reactivation/reinfection with herpesviruses in immunocompromised patients is a common finding after intensive anticancer therapy. Therefore, cytomegalovirus (CMV), Epstein–Barr virus (EBV), human herpesvirus type 6 (HHV6), and herpes simplex virus (HSV) are often registered at sufficient rates in the blood and other biological samples within the first 100 days post-HSCT.

The major diagnostic issue is to discern between immune-mediated lesions typical of intestinal GVHD and potential viral lesions of the target tissues. Histological evaluation of colonic biopsies is a standard technique of intestinal GVHD assessment [4]. A search for potentially infectious pathogens may be also performed in mucosal biopsies [5,6]. However, the role of viruses in gastrointestinal complications after HSCT is still in question. Detection of viral antigens and/or DNA in the intestinal biopsies is routinely performed with immunological and PCR techniques [7,8].

To our knowledge, the detection rates and timing of herpesvirus activation have not yet been evaluated in gut mucosal biopsies taken in cases of severe intestinal GVHD after HSCT, despite the need for improved diagnosis of virus-related disorders. The aim of present study was to assess the time dependence of the incidence of the four most common herpesviruses detected in the gut mucosa following allogeneic HSCT. We revealed the time dependence of EBV and CMV detection, as well as a correlation between the incidence of EBV and hematopoietic recovery post-transplant.

## 2. Materials and Methods

### 2.1. Ethical Statement

This study was approved by the local Institutional Review Board at the First St. Petersburg State I. Pavlov Medical University (protocol number 214 of 17 December 2018). All endoscopic procedures and methods of collecting biological samples used under this protocol were taken only with the approval of the attending physician. The study was conducted in accordance with guidelines of the 1964 Declaration of Helsinki and its later amendments. All patients or their guardians signed a written informed consent form for a hematopoietic stem cell transplant and the subsequent medical procedures, as well as potential usage of their clinical data for the purposes of clinical research.

### 2.2. Patients

Our study group included 119 patients admitted to the R. Gorbacheva Memorial Research Institute of Pediatric Oncology, Hematology, and Transplantation for allogeneic hematopoietic stem cell transplantation (allo-HSCT) between January 2015 and December 2020. Most patients were treated for acute myeloid leukemia (AML, *n* = 56) or acute lymphoblastic leukemia (ALL, *n* = 54). Within 6 months post-transplant, along with effective immunosuppressive therapy, these patients underwent intensive care treatment due to severe gastrointestinal complications of infectious origin, or acute or chronic graft-versus-host disease (GVHD). The main demographic and clinical characteristics of the patients are presented in Table 1.

The severity of oral and intestinal mucositis was evaluated by the WHO criteria, from erythema (Grade 1) to disabling mucosal ulcers (Grade 4). The severity of intestinal syndrome was assessed by means of the 7-point Bristol scale [9]. Evaluation of pain symptoms was performed by a visual analog 10-point scale (VAS). Acute GVHD grade was assessed according to the standard scale [10]. Liver GVHD was graded according to the biochemical markers of hepatic disorders (serum bilirubin, transaminases etc.).

### 2.3. Endoscopic Sampling and General Examination

In total, 155 diagnostic biopsies of large bowel mucosa were performed within 180 days after allo-HSCT by appropriate clinical indications in cases of acute intestinal GVHD resistant to steroid therapy and other conventional treatments, or in severe intestinal syndrome associated with *C. difficile* or verified *Klebsiella* infection. To perform fibrocolonoscopy and diagnostic biopsies, we used the CF Q 150 L Olympus (for children), EC-3890LK Pentax, or FC-1Z Fujinon devices. In special cases, patients with recurrent intestinal syndrome were subjected to repeated diagnostic intestinal biopsies in order to verify GVHD and/or infectious lesions of the mucosal tissues. Most biopsies were mostly taken from the descending and sigmoid colon (52%), transverse colon (25%), ascending colon (10%), caecum (5%), and ileum (5%). Rectal samples were taken in 3% of the cases. The mucosal samples (3 to 6 specimens) were taken by means of forceps biopsy during the endoscopic procedure. Part of the biomaterial was subjected to common histology of paraffin-embedded tissue samples. This was performed by standard techniques using routine H&E staining and immunohistochemical studies of CD3+ cells and viral antigens, if required. The HSCT patients underwent routine physical examinations; blood counts and blood biochemistry had been monitored in the early stages post-transplant and during intensive care therapy. Quantitative testing for serum IgG antibodies against CMV, EBV, and HSV was performed by the ELISA technique using an Evolis automatic analyzer (Stratec Biomedical Systems AG, Birkenfeld, Germany), being positive in the majority of cases.

### 2.4. PCR Diagnostics

To perform viral PCR diagnostics, the biopsy samples were placed into physiological saline and immediately delivered to the Department of Clinical Microbiology. The samples were rinsed in a lysis solution (DNA-Technology, Moscow, Russia) and subjected to a brief ultrasonic treatment until homogenization of the sample. DNA was extracted with DNA-Sorb isolation kits (DNA-Technology, Moscow, Russia). PCR detection of the virus-specific DNA was performed with commercial PCR kits for qualitative gene-specific diagnostics (InterLabservis, Moscow, Russia). The PCR products were detected by routine gel electrophoresis in 1.5% agarose stained with ethidium bromide. The results were considered positive or negative, presuming that the sensitivity of the commercial test kits was, respectively, 400, 400, 1000, and 400 gene copies per mL of tissue lysate for EBV, CMV, HSV (types 1 + 2), and HHV type 6 A/B. These cutoff values were reported by the manufacturer of the test kits. Moreover, additional quantitative assays were performed in our laboratory, with commercial real-time PCR kits from the same firm using Bio-Rad IQ or CFX96 thermocyclers. Positive findings of any virus were regarded as its activation in the given tissue sample.

### 2.5. Statistics

Statistical evaluation was performed by Statistica 10 software, using parametric (Student’s *t*-test) and non-parametric criteria (Chi-square test, Spearman’s correlation criterion). The level of confidence was taken as *p*
< 0.05.

## 3. Results

### 3.1. Relative Frequency of Common Herpesviruses in the Tissue Biopsies

In general, the mean detection frequencies for the four clinically relevant viruses (CMV, EBV, HSV, and HHV6) in the biopsy samples proved to be stable over time; however, this depended on the viral species. For example, HSV positivity was rather infrequent in colon mucosa, with maximal levels of 8% at 2–3 months post-transplant (Figure 1). By contrast, high incidence was observed for HHV6 (55 to 75%, with a mean of 62.4%) during 6 months after HSCT. Cytomegalovirus DNA was also rather common in mucosal biopsies (28–35%) during 1 to 4 months post-transplant (Figure 2). The EBV incidence in colonic samples exhibited significant variability during the post-HSCT period, with significant increases in later periods (~35% at 5–6 months post-HSCT, *p* = 0.03). Therefore, we chose HHV6, CMV, and EBV for subsequent clinical association studies.

Of special interest was a comparison between the incidence of herpesviruses in colonic biopsies and time-matched blood samples from the same patients. In total, 30 time-compatible blood–biopsy pairs were available. In summary, the frequency of HHV6 in colon samples (59%) was sufficiently higher than in time-matched blood samples from these patients (18%, *p* < 0.04). A significant correlation was found between HHV6 incidence in blood and intestinal samples (r = 0.451; *p* = 0.02). CMV detection rates in colonic biopsies were similar to the associated blood samples, i.e., 33.3% and 25.8%, respectively (*n* = 30). A significant correlation was found between CMV incidence in blood and colonic samples (r = 0.489; *n* = 82; *p* = 0.03). The mean EBV incidence in colonic biopsies was also similar to the time-matched blood samples, i.e., 18.5% and 19.1%. However, a sufficient correlation was found between EBV incidence in blood and intestinal samples (r = 0.583, *p* = 0.003).

### 3.2. Incidence of Epstein–Barr Virus and Post-Transplant Complications

Looking for possible associations between the positive viral findings in mucosal biopsies and the clinical characteristics of the patients, we did not reveal any significant correlations for most demographic and clinical features of the patients subjected to HSCT, including age, sex, or primary oncological diagnosis. However, we revealed some associations with post-transplant hematopoietic reconstitution (see Table 2). The incidence of HHV6 and CMV in colonic biopsies did not show any significant associations with post-transplant leukocyte recovery, oral mucositis, skin, or advanced-grade intestinal GVHD.

In addition, the increased EBV incidence was found to be associated with a number of HSCT parameters and post-transplant complications. In the early stages post-transplant, the incidence of Epstein–Barr virus in gut biopsies was similar for different biopsy sites. Notably, a higher EBV incidence was registered in the patients transplanted with bone marrow versus peripheral blood stem cells (28% versus 11.6%, *p* = 0.015). Prolonged terms of post-transplant blood cell reconstitution in EBV-positive cases have been shown for blood platelet counts (*p* = 0.023) and, as a trend, for blood leukocytes (*p* = 0.06). Of note, significantly increased rates of EBV detection in colonic samples were revealed in later periods, with a peak in the fifth month after HSCT (*p* < 0.05), thus reflecting possible EBV activation in large bowel mucosa in this period post-transplant.

Finally, we found an increased percentage of EBV-positive biopsies among the patients who did not survive the severe intestinal syndrome despite intensive care measures, i.e., 56% (nine cases) compared with 21% in the survivors (44 cases, *p* = 0.02). This difference was not registered for CMV, HSV, or HHV-6, thus suggesting a special role of EBV in post-transplant complications.

## 4. Discussion

In our study, large differences in frequency were revealed for the four clinically established herpesviruses in the gastrointestinal mucosa following HSCT and the subsequent immune deficiency. For example, herpes simplex virus activation was uncommon, despite its known tropism for the oral and mucosal epithelium. The low HSV incidence in our patients may be due to acyclovir prophylaxis already having been used over the last 15 years in current protocols of intensive chemotherapy/HSCT.

Studies of local CMV reactivation in the patients with intestinal disorders were performed during the last decades using quantitative and qualitative evaluations of the viral loads. For example, Ganzenmueller et al. [11] measured CMV gene copy numbers by quantitative real-time PCR in 163 colonic biopsies in immunocompromised patients. The authors showed CMV in a high proportion (32.5%) of the biopsies, though with broad scatter (>6 log_10_ range of CMV copy numbers per cell). The biopsies from the patients with clinically and pathologically confirmed CMV intestinal disease showed a significantly higher CMV load (*p* < 0.001). In general, the results of quantitative PCR, while being more precise, still provide a high scatter of the data, thus preventing optimal clinical conclusions and statistical issues. Therefore, most published studies on viral reactivation are ultimately based on qualitative PCR findings, with defined sensitivity thresholds. Hence, we considered positive tests with established cutoff levels to be activation markers of viral infection.

Sufficient incidence rates of CMV and EBV were found in colonic samples of the HSCT patients with intestinal complications of inflammatory origin (colitis or GVHD). These results are in accordance with several studies performed in patients with inflammatory bowel disease (IBD). For example, a higher prevalence of EBV was found in more severe cases rather than in the controls and mild clinical cases [8,12]. Interestingly, this dependence of the viral loads of EBV and CMV on the clinical severity of IBD was also found in peripheral blood samples. These results are in agreement with the similar correlation in EBV positivity between blood and colonic samples revealed in our patients despite the different origin of their inflammatory conditions.

Moreover, we found significant associations between a higher Epstein–Barr virus (EBV) frequency in the mucosal samples and the usage of bone marrow. Hematopoietic stem cells from the bone marrow are well known for their slower regeneration than peripheral stem cells, thus causing prolonged leukocyte and platelet recovery post-transplant. In fact, a higher EBV incidence in colonic mucosa was registered in patients with longer leukocyte and platelet reconstitution. This finding agrees with previous evidence of delayed leukocyte reconstitution, such as the recovery of B and T cells at >90 d post-transplant in HSCT patients with a high EBV content in the blood plasma (>1000 gene copies/mL) versus the comparison group [13]. Therefore, the activation of EBV in colonic samples may be a marker of suppressed immunity, thus promoting local complications of immune and infectious origin.

The significant role of EBV in the genesis of lymphoid gastrointestinal neoplasia has been suggested over many years [14]. Generally, lymphoid cells are typical hosts for latent and replicating EBV. Important observations were made by Ryan et al. [15]. The workers searched for EBV markers in archived, paraffin-embedded gastrointestinal biopsies from patients with gastritis and IBD using a comprehensive panel of PCR and immunohistochemical techniques. EBV DNA that was detectable by different DNA markers was detected in up to 64% of samples from IBD patients. Meanwhile, expression of the specific EBER antigen was shown to be associated with tissue-infiltrating CD20+ lymphocytes, thus being a possible source of the detectable virus in the inflamed colonic tissue. The EBV-harboring lymphocytes may be a similar source of the detectable virus in colonic specimens following HSCT. GVHD is an acute inflammatory condition occurring because of the aggressive effects of donor lymphocytes upon the skin and intestinal epithelial cells of the patient [1]. An association between EBV positivity in the gut mucosa and skin GVHD in our patients is in line with the suggested infiltration of EBV-infected lymphocytes into the inflamed epithelial tissue post-transplant.

Distinct interrelations among EBV burden, proinflammatory interleukins in blood samples, and acute GVHD after HSCT in children were revealed in a recent clinical study [16]. In general, the interplay of EBV activation, the gut microbiome, and the resulting inflammatory responses is still unclear, as shown in the study performed in HIV-infected patients [17]. However, possible role of EBV reactivation/reinfection in GVHD pathogenesis has been suggested by several works. For example, comprehensive surveys [18,19] performed in >11,000 HSCT recipients with leukemia have shown that the HSCTs from EBV-seropositive donors displayed a higher risk of acute and, especially, chronic GVHD than those from seronegative donors. A similar association of long-term EBV reactivation and chronic GVHD was revealed by Japanese researchers [20].

Therefore, more complex studies are required to evaluate the associations among intestinal syndromes, local immunity, and viral reactivation in the immune complications developing after HSCT. More extensive studies of the intestinal virome following HSCT may be performed by novel metagenomic approaches [21]. The authors detected multiple virus-specific DNA sequences in the stool samples of HSCT recipients and revealed higher detection rates, e.g., for herpesviruses, papillomaviruses, and polyomaviruses in patients with enteric GVHD. Future studies of stool and intestinal biopsies should allow more insight into the diagnostics and pathogenic role of intestinal infectious agents during the post-transplant period [22].

## 5. Conclusions

The four clinically relevant herpesviruses were detectable in the colonic mucosa during 6 months after hematopoietic stem cell transplantation, with highest incidence rates for HHV6, followed by CMV, EBV, and HSV.

A significant correlation was found between the incidence of EBV in blood and gut biopsies.

Higher EBV detection rates in colonic mucosal biopsies were associated with the usage of bone marrow grafts and with the delayed recovery of leukocytes and platelets post-transplant.

An increased frequency of EBV-positive colonic biopsies was found in deceased patients with intestinal syndromes after HSCT.

## Figures and Tables

**Figure 1 microorganisms-10-02128-f001:**
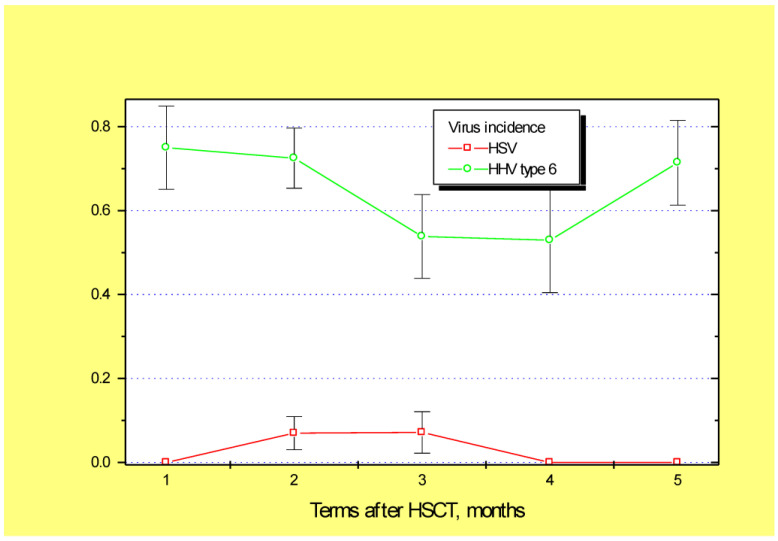
General incidence of HSV (red squares) and HHV-6 (green circles) in the intestinal biopsies from the first month (D + 1 to D + 30) up to 5–6 months after allogeneic HSCT. Abscissa, time period following HSCT, in months; ordinate, incidence of positive PCR viral tests (M ± m).

**Figure 2 microorganisms-10-02128-f002:**
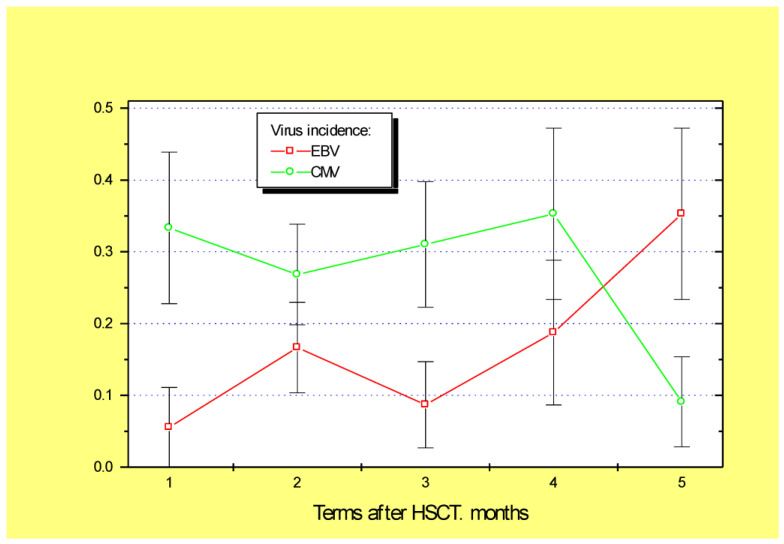
Incidence of EBV (red squares) and CMV (green circles) in the intestinal biopsies from the first month (D + 1 to D + 30) up to 5–6 months after allogeneic HSCT. Abscissa, time period following HSCT, in months; ordinate, incidence of positive PCR viral tests (M ± m).

**Table 1 microorganisms-10-02128-t001:** Clinical characteristics and HSCT parameters of post-transplant patients with intestinal syndrome.

Demographic Data, Primary Diagnosis		Parameters of Hematopoietic Transplantation	
Number of patients	119	HSCT mode:	
Mean age ± SD	28.3 ± 19.0	Haploidentical related	48.2%
Median age	24 (1–72)	Allogeneic unrelated	40.8%
Sex, males females	56% 44%	Allogeneic related	11.0%
Body mass index ± SD	23.4 ± 12.5	CD34+ cells/kg (M ± SD)	5.9 + 3.0 (0.7–17; Med. 6)
Primary clinical diagnosis:	Number of cases:	Stem cell source:	
Acute myeloid leukemia	34	Bone marrow	41.6%
Acute lymphoblastic leukemia	36	Peripheral blood stem cells	58.4%
Chronic myeloid leukemia	8	Oral mucositis, grade	0: 67.1% 1–2: 19.7% 3–4: 13.2%
Severe aplastic anemia	15		
Hodgkin’s disease	10		
Non-Hodgkin’s lymphoma	7		
Other disorders	9		
Herpesvirus-positive biopsies	27.7%	Skin GVHD grade	0: 35.6% 1: 12.4% 2: 21.7% 3: 30.3%
CMV	25.8%		
EBV	18.7%		
HSV	3.9%		
HHV type 6	62.4%		
Conditioning chemotherapy: Myeloablative	56.5%	Intestinal GVHD grade	0: 41.7% 1: 5.3% 2: 15.4% 3–4: 37.6%
Non-myeloablative	43.5%		

**Table 2 microorganisms-10-02128-t002:** Relative frequency of the three herpesviruses in the series of large bowel biopsies: dependence on the parameters and complications of HSCT (in parentheses, number of samples).

HSCT Parameters	HHV6-Positive Samples, %	*p* Levels	CMV-Positive Samples, %	*p* Levels	EBV Positive Samples, %	*p* Levels
PCR positivity with different HSC sources:						
Bone marrow	65.6% (*n* = 40/61)	0.564	16.1% (*n* = 17/62)	0.737	28.1% (*n* = 16/57)	**0.015**
Peripheral stem cells	60.9% (*n* = 57/94)		23.9% (*n* = 23/92)		11.6% (*n* = 9/78)	
CD34+ stem cells/kg:						
Virus-positive	5.91 ± 0.31 (93)	0.818	6.12 ± 0.48 (40)	0.562	5.97 ± 0.54	0.675
Virus-negative	5.94 ± 0.42 (55)		5.7 ± 0.28 (113)		5.89 ± 0.29	
Leukocyte recovery, d:						
Virus-positive	21.01 + 0.80 (87)	0.362	19.29 ± 0.91 (38)	0.431	22.04 + 1.17	**0.06**
Virus-negative	19.56 + 0.91 (53)		20.94 ± 0.73 (109)		20.06 + 0.73	
Platelet recovery (>20/µL), days						
Virus-positive	24.71 + 2.28 (74)	0.195	19.27 + 2.22 (33)	0.381	24.77 + 3.18 (22)	**0.023**
Virus-negative	18.98 + 1.87 (46)		23.83 + 1.90 (94)		21.24 + 1.76 (86)	
**Clinical complications:**						
Virus-positive, oral mucositis,						
Grade 0	65.0% (26/40)	0.369	31.0% (13/42)	0.07	25.7% (9/35)	0.76
Grade I–II	75.0% (9/12)		36.3% (4/11)		18.2% (2/11)	
Grade III–IV	87.5.0% (7/8)		25.0% (2/8)		42.9% (3/7)	
Skin GVHD, virus-positive tests, %						
Grade 0	71.7% (33/46)	0.140	34.8% (16/46)	0.275	7.3% (3/41)	**0.04**
Grade I–II	50% (21/42)		17.0 % (8/47)		30.0% (15/40)	
Grade III–IV	68.4% (26/38)		28.2% (11/39)		19.4% (7/36)	
Intestinal GVHD, virus-positive tests, %						
Grade 0	65.4% (34/52)	0.127	30.9% (17/55)	0.286	18.4% (9/49)	0.12
Grade I–II	48.0% (12/25)		14.5% (4/28)		4.3% (1/23)	
Grade III–IV	69.4% (34/49)		28.6 (14/49)		26.7% (12/45)	

## Data Availability

We report no links to publicly archived datasets analyzed or generated during the study.

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
