# Peer review of "Incidence of Common Herpesviruses in Colonic Mucosal Biopsies Following Hematopoietic Stem Cell Transplantation"

_microorganisms, 2022, doi:10.3390/microorganisms10112128_

Round 1
Reviewer 1 Report
This study provided sufficient experimental evidence to support the conclusion.
Overall the results are solid and consistent.
However, as specified in the comments to the authors, some important previous studies are not mentioned, and the manuscript needs careful editing, ideally by a native English speaker. Thus, substantial revision seems required before publication.
Author Response
Dear Reviewer,
Much thanks for reading and frank evaluation of the article.
I would like to answer You remarks:
- Important previous studies are not mentioned: the role of EBV as a factor of bowel inflammation and its affinity for lymphocytes is described in 3 additional works which are discussed and cited (Discussion, page 7, para 3 and 5; refs No. 12, 13, 16).
- The text needs editing by native English: the main text was carefully read and sufficiently edited by a specialist in this area with fluent English.
Reviewer 2 Report
The aim of this study was to assess time-dependence for the incidence of 4 dominant herpesviruses detected in gut mucosa following allogeneic HSCT. They have revealed a time dependence of EBV and CMV detection, as well as correlation between EBV incidence and hematopoietic recovery post-transplant.
Two questions:
1. page 7,line 214-216,please reconstruct the sentence.
2. Page 4, line 117-128, the readers will be difficult to understand the results of the qualitative PCR, do you need to do electrophoresis analysis?
Author Response
Dear Reviewer,
Much thanks for careful reading and providing a review of our manuscript.
My reply to Your remarks is as following (by original version):
- page 7, line 214-216 - please, reconstruct the sentence ...Therefore, one may suggest that EBV presence in colonic samples may be a factor of impaired immunity causing local complications of immune and infectious origin, e.g., bacterial infections... It is now replaced by .... Therefore, EBV activation in colonic samples may be a marker of suppressed immunity, thus promoting local complications of immune and infectious origin (revised versoin, Discussion, paragraph 4, bottom lines).
- page 4, line 117-128, the readers will be difficult to understand the results of quantitative PCR, do you need to do electrophoresis analysis? We have added more details of PCR testing, both quantitative and qualitative (electrophoresis) as shown in page 4, section 2.4., lanes 7-14 (in revised version).